# Pericyte Loss in Diseases

**DOI:** 10.3390/cells12151931

**Published:** 2023-07-26

**Authors:** Pengfei Li, Hongkuan Fan

**Affiliations:** Department of Pathology and Laboratory Medicine, Medical University of South Carolina, Charleston, SC 29425, USA; lippe@musc.edu

**Keywords:** perictyes, vascular leak, diabetes, blood–brain barrier

## Abstract

Pericytes are specialized cells located in close proximity to endothelial cells within the microvasculature. They play a crucial role in regulating blood flow, stabilizing vessel walls, and maintaining the integrity of the blood–brain barrier. The loss of pericytes has been associated with the development and progression of various diseases, such as diabetes, Alzheimer’s disease, sepsis, stroke, and traumatic brain injury. This review examines the detection of pericyte loss in different diseases, explores the methods employed to assess pericyte coverage, and elucidates the potential mechanisms contributing to pericyte loss in these pathological conditions. Additionally, current therapeutic strategies targeting pericytes are discussed, along with potential future interventions aimed at preserving pericyte function and promoting disease mitigation.

## 1. Introduction

Pericytes were first characterized by Eberth and Rouget nearly 150 years ago [1,2,3]. They are specialized cells located within the capillary basement membrane that wrap around endothelial cells in the microcirculation throughout the body [4]. Although a unique pericyte marker has yet to be identified, pericytes are commonly characterized by a combination of markers, including platelet-derived growth factor receptor β (PDGFRβ), the proteoglycan neural glial antigen-2 (NG2), alpha-smooth muscle actin (α-SMA), CD13, CD73, and CD146 [1,2,5,6,7]. It should be noted that the expression of these markers on pericytes is dynamic and tissue dependent [2,8]. Consequently, the distribution and function of pericytes are associated with endothelial barrier properties and exhibit tissue-specific variations [4,8] (Table 1). The developmental origin of pericytes is heterogeneous and remains largely unknown [9]. The origin of pericytes in the lung and liver can be traced to the mesothelium, while the origin of pericytes in the heart was traced to the epicardial mesothelium [9]. The brain and retina harbor the highest density of pericytes, with a pericyte-to-endothelial cell (EC) ratio of 1:1 [2,4,8,10]. These pericytes play a critical role in establishing the blood–brain barrier (BBB) and blood–retinal barrier (BRB), safeguarding brain and retinal cells against potentially harmful blood-derived factors [2,4,8,10]. The typical pericytes in the central nervous system (CNS) are flattened, or elongated, stellate-shaped solitary cells [11]. In the lung, pericytes are indispensable for maintaining pulmonary vasculature function and optimal gas exchange, with a pericyte-to-EC ratio of approximately 1:7–1:9 [8,12]. These cells have a spindle-shaped or stellate morphology and elongated, multibranching cellular processes [13]. However, activated and differentiated lung pericytes also contribute to inflammatory responses and fibrosis [14,15,16]. Renal pericytes, including peritubular pericytes, mesangial cells, and podocytes (pericyte-like cells), are involved in regulating blood ultrafiltration and vascular permeability, with a pericyte-to-EC ratio of about 1:2.5 in the kidney [2,4,17]. Mesangial cells are rounded and compact [11]. Hepatic stellate cells (HSC), the pericytes in the liver, are located between the parenchymal cell plates and the sinusoidal endothelial cells. These cells are characterized by their dendritic structures with cytoplasm filled with fat-storing droplets containing vitamin A and are implicated in extracellular matrix remodeling, vitamin A metabolism, and the recruitment of inflammatory cells [2,18,19,20]. In the liver, the ratio of pericytes to ECs is approximately 1:10 [8,18]. In the heart, pericytes show typical spindle-shaped morphology [21]. The coverage of cardiac pericytes to endothelial cells is about 1:2 to 1:3, and they play a critical role in regulating blood flow, vascular remodeling, and maintaining vascular function [22]. Collectively, pericytes contribute to essential functions in various organs, and alterations in pericyte coverage or density can lead to vascular disturbances and organ dysfunction.

Increasing evidence supports the notion that pericyte loss disrupts vascular homeostasis and contributes to disease progression in diverse conditions, including diabetes, Alzheimer’s disease (AD), amyotrophic lateral sclerosis (ALS), sepsis, stroke, and traumatic brain injury (TBI). This article aims to provide a comprehensive summary of the role and mechanisms of pericyte loss and discuss potential therapies targeting pericytes.

## 2. Pericyte Loss in Diseases

### 2.1. Pericyte Loss in Diabetes

Diabetes is a chronic health condition that causes multiple vascular complications, such as retinopathy and nephropathy [23]. Pericyte loss is an early hallmark of diabetes-associated microvascular diseases and plays a crucial role in the disease progression of various organs including the retina, kidney, brain, and heart [23,24].

#### 2.1.1. Pericyte Loss in Diabetic Retinopathy (DR)

DR, a major complication of diabetes, is the leading cause of blindness worldwide and is characterized by vascular damage in the retina [25,26,27]. Among the vascular cells, pericytes are the earliest to be affected by diabetes, and their loss is a hallmark of diabetic retinopathy, contributing to blood vessel leakage [28]. The loss of pericytes has been detected in the retinas of diabetic patients [29,30,31], as well as in various animal models, including mice [31,32,33], dogs [34,35], hamsters [36], and rats [37,38].

The precise mechanisms underlying pericyte loss in diabetic retinopathy are not yet fully understood. One hypothesis suggests that pericyte death via apoptosis is involved, as confirmed by studies in both patients and animal models of DR [30,32,39,40,41,42]. Mechanically, factors such as high-glucose [40,43], oxidative stress [44], advanced glycation end products [45,46], TNF-α [47], and IL-1β [48] have been shown to induce apoptosis in retinal pericytes in vitro and/or in vivo. Additionally, the migration of retinal pericytes may contribute to pericyte loss in DR, regulated by the Ang-Tie system and autophagy processes [49,50]. Pericyte loss contributes to EC-pericyte dissociation and vascular dysfunction as retinal capillary pericytes are critical to maintaining EC-pericyte contacts and the integrity of vascular barrier function via secretion of sphingosine 1-phosphate [51,52]. Therefore, preventing the loss of retinal pericytes would be beneficial. Pericytes derived from adipose-derived stem cells (ASCs) showed protective effects against capillary loss in the retina in a murine model of DR [53]. Additionally, human pericyte-like ASCs have demonstrated the ability to protect human retinal endothelial cells in an in vitro model of DR [54]. Therefore, pericyte/pericyte-like cell-targeting therapies or cell implantation of pericytes/pericyte-like cells hold promise for the treatment of DR and warrant further investigation.

#### 2.1.2. Pericyte Loss in Diabetic Nephropathy (DN)

As a common complication of diabetes, DN is characterized by proteinuria, microvascular damage, and the disruption of glomeruli and the tubular system [4,23,55,56]. It significantly impacts the quality of life of diabetic patients and is the leading cause of end-stage renal disease [23,56]. Renal pericyte-like cells, including peritubular pericytes, mesangial cells, and podocytes, are susceptible to oxidative stress induced by high glucose and play a critical role in the progression of DN [4,23]. 

Several hallmark features of DN, such as peritubular capillary rarefaction (PTC) and peritubular fibrosis, mesangial and glomerular hypertrophy, and podocyte injury, are closely associated with renal dysfunction [4,55,57,58,59]. Peritubular pericytes are crucial for maintaining the integrity of peritubular capillaries, as the loss of pericytes can accelerate PTC rarefaction [58,60,61]. Furthermore, the migration of peritubular pericytes away from the capillaries and their transformation into myofibroblasts are potential mechanisms underlying PTC and peritubular fibrosis [4,62,63]. However, the precise role and underlying mechanisms of peritubular pericytes in DN remain largely unknown. Mesangial cells, comprising approximately 30% of glomerular cells, undergo hypertrophy in the early stages of DN [4,23,64,65]. Increased mTOR activity may contribute to mesangial cell hypertrophy under high glucose conditions [4,65]. Moreover, diabetic rats and mice exhibit glomerular cell loss and apoptosis, which are associated with albuminuria and renal dysfunction [66,67]. Mechanistically, elevated levels of urinary miR-15b-5p have been observed in diabetic patients and db/db mice and contribute to high glucose-induced mesangial cell apoptosis [68]. Additionally, serum levels of Angpt2 were increased in diabetic patients and db/db mice, and the Angpt2/miR-33-5p/SOCS5 signaling pathway has been implicated in mesangial cell apoptosis under high glucose conditions [69]. 

Podocytes, which are pericyte-like cells, play a crucial role in the progression of DN [4,23,70], and podocyte injury is a hallmark of both DN and non-diabetic kidney diseases [71,72]. The loss of podocytes serves as an early pathological marker and contributes to proteinuric glomerulopathies in DN [70,71,73]. The number of podocytes is decreased in both type 1 and type 2 diabetic patients [74,75], and podocyte loss has also been observed in diabetic mice [76,77] and rats [78,79]. Podocyte apoptosis is the most common mechanism of podocyte loss and has been extensively documented [71,80,81,82]. Mechanistically, the accumulation of harmful factors such as reactive oxygen species (ROS), advanced glycation end products, miRNAs, and angiotensin II, along with the activation of signaling pathways, including p53, mTOR, and Notch, are involved and contribute to DN-induced podocyte apoptosis [71,72,83]. Several other pathways have been implicated in podocyte loss in DN, including autophagy, mitotic catastrophe, anoikis, necrosis, and pyroptosis, which have been comprehensively discussed by Jiang et al. [71]. VEGF-A, primarily produced by podocytes, is necessary for the survival of glomerular endothelial cells [84,85]. The loss of podocyte-derived VEGF-A results in EC dysfunction and disrupts the glomerular filtration barrier [84,86]. Therefore, podocyte loss has a significant impact on the dysfunction of EC and the glomerular filtration barrier. 

#### 2.1.3. Brain Pericyte Loss in Diabetes

Diabetes can induce damage that leads to dysfunction of the BBB and cognitive decline in both patients and experimental models [87,88,89,90]. Furthermore, diabetic patients have a higher risk of developing dementia-related diseases such as stroke and Alzheimer’s disease (AD) [91,92]. In the brain, diabetes-related complications are characterized by pericyte loss, increased BBB permeability, and neuronal dysfunction [87,89,93]. Reduced numbers of brain pericytes have been reported in diabetic patients [23,94], as well as in animal models of diabetes, including mice and rats [95,96,97]. In vitro studies have shown that oxidative stress induced by high glucose can lead to apoptosis in cultured brain pericytes [98,99,100,101]. The activity of mitochondrial carbonic anhydrases was believed to induce brain pericyte loss in diabetic mice as the inhibition of mitochondrial carbonic anhydrases activity can reduce oxidative stress and prevent pericyte dropout [96]. However, the exact mechanisms and in vivo processes underlying brain pericyte loss in diabetes require further investigation.

#### 2.1.4. Cardiac Pericyte Loss in Diabetes 

Cardiovascular disease (CVD) is a significant complication of diabetes and is the leading cause of heart failure or mortality in diabetic patients [102,103,104,105]. Pericytes play a crucial role, particularly in the early stages of diabetes-associated CVD, including myocardial and interstitial fibrosis [105,106,107]. The loss of pericytes has been demonstrated in the hearts of diabetic patients and diabetic pigs [108]. Additionally, studies by Tu et al. showed a reduction in the number of cardiac pericytes and microvascular coverage in diabetic mice [24]. The overexpression of thymosin beta 4 has the ability to mitigate cardiac pericyte loss in diabetic pigs, providing a potential therapeutic approach for diabetes-associated CVD [108]. However, the specific underlying mechanisms of cardiac pericyte loss in diabetes remain unclear and require further investigation.

In summary, pericyte loss is closely associated with various complications of diabetes and significantly contributes to disease development. Further research is needed to gain a better understanding of the underlying mechanisms involved and to explore novel therapeutic strategies targeting pericytes.

### 2.2. Pericyte Loss in Aging and Neurodegenerative Diseases 

#### 2.2.1. Pericyte Loss in Alzheimer’s Disease

AD is the most prevalent neurodegenerative disorder characterized by cognitive impairment, an accumulation of amyloid β-peptide (Aβ), BBB dysfunction, and neuroinflammation [109,110,111]. Pericytes play a critical role in AD, as their deficiency in mouse models of AD accelerates BBB breakdown and increases Aβ accumulation in the brain [112]. Pericyte loss has been reported in various regions of AD patients’ brains, including the white matter [113,114], precuneus [115], cortex [112,116], hippocampus [7,116], and retina [117]. Similarly, reduced pericyte numbers have been observed in the cortex [118,119], hippocampus [7,119], and retina [120] of AD mice. In the retina, the activation of inflammation appears to contribute to pericyte loss as an association between NF-κB p65 phosphorylation levels and vascular PDGFRβ expression was observed in AD mice [120]. Apoptosis is believed to contribute to pericyte loss, as pericyte apoptosis has been identified in the retina and hippocampus of AD patients [7,117]. In vitro studies have shown that Aβ stimulation induces apoptosis in cultured brain pericytes [7,119]. Mechanistically, decreased miR-181a levels and enhanced Fli-1 expression may contribute to pericyte loss and apoptosis in AD [7,119]. A reduced miR-181a expression has been observed in AD mice, but the overexpression of miR-181a can mitigate pericyte loss, improve BBB function, and decrease Aβ accumulation [119]. Furthermore, miR-181a inhibits Aβ-induced pericyte apoptosis in murine brain cell cultures [119]. Our recent study suggests that Fli-1 expression is increased in postmortem brains from AD donors and in a mouse model of AD known as 5xFAD. The inhibition of Fli-1 via antisense oligonucleotide Fli-1 Gapmer decelerates pericyte loss, reduces inflammatory response, ameliorates cognitive deficits, improves BBB function, and decreases Aβ deposition [7]. In addition, Fli-1 Gapmer treatment protects against Aβ-induced apoptosis in human brain pericytes in vitro [7]. Aβ-evoked pericyte-mediated constriction of the cerebral capillary bed contributes to the reduction in cerebral blood flow during AD [121]. Moreover, EC-pericyte contacts are important to control cerebral blood flow and promote EC survival via pericyte-derived VEGF [122]. The loss of pericytes leads to increased VEGF expression in EC, which may be a compensatory signaling pathway [123]. Thus, pericyte loss may contribute to reduced cerebral blood flow and EC dysfunction. The implantation of pericytes derived from mesenchymal stem cells has been shown to enhance cerebral blood flow and reduce Aβ levels in a mouse model of AD, suggesting that cell-based therapies targeting pericytes/pericyte-like cells may hold promise in the prevention and treatment of AD [124].

#### 2.2.2. Pericyte Loss in Amyotrophic Lateral Sclerosis (ALS)

ALS, a fatal neurodegenerative disorder, is characterized by blood–spinal cord barrier dysfunction and the progressive degeneration of motor neurons [125,126,127,128]. Recent studies have highlighted the important role of pericytes in ALS [129]. Decreased pericyte coverage or number has been observed in the ventral horn and spinal cord of ALS patients, which correlates with vascular disruption [130,131]. Furthermore, a loss of pericytes in the choroid plexus has been detected in patients with ALS, coupled with a deregulation of the blood–cerebrospinal fluid (CSF) barrier [132]. In a murine model of ALS, reduced pericyte coverage in spinal cord capillaries has also been demonstrated [133]. Interestingly, the administration of adipose-derived pericytes has shown promising results in ALS mice, extending their survival and increasing antioxidant enzymes in the brain [134]. These findings suggest that pericytes may represent a novel potential cell therapy for treating ALS, although further studies are needed to fully understand pericyte loss in ALS and its implications for disease progression.

Overall, pericyte loss in aging and neurodegenerative diseases poses a significant challenge that can have negative effects on brain health. Advancing our understanding of the underlying mechanisms of pericyte loss and developing new treatments to prevent or reverse this process are important areas of future research.

### 2.3. Pericyte Loss in Infectious Diseases

#### 2.3.1. Pericyte Loss in Sepsis

Sepsis is a life-threatening condition caused by a microbial infection resulting in organ dysfunction and failure. It is characterized by a systemic inflammatory response and microvascular dysfunction [135,136]. Recent studies have highlighted the role of dysfunctional pericytes in sepsis-induced microvascular dysfunction, which serves as a hallmark of severe sepsis and septic shock [15,137]. Research by Nishioku et al. demonstrated the detachment of pericytes from the basal lamina in the hippocampus of LPS-treated mice [138]. The detachment of pericytes may contribute to sepsis-induced BBB dysfunction [139] as pericytes control vascular permeability in the brain [140]. Pericyte loss has also been observed in the lungs and hearts of LPS-treated mice, although this loss is not caused by apoptosis [141]. Reduced pericyte coverage in mesenteric microvessels has been demonstrated in both cecal ligation and puncture (CLP) and LPS-induced septic rats [142]. In a previous study, we showed a reduction in pericyte density in the lungs and kidneys of CLP-induced septic mice, suggesting pericyte pyroptosis as a potential mechanism for this loss [15]. Our findings indicate that an increased expression of Fli-1 in lung pericytes may contribute to pericyte pyroptosis and the knockout of Fli-1 in pericytes attenuates lung pericyte loss, vascular leak, and mortality in a murine model of sepsis [15]. Moreover, angiopoietin-2, which is increased in septic patients, has been implicated in pericyte loss, as endothelial angiopoietin-2 overexpressed mice displayed significant pericyte loss [143,144]. Furthermore, the disruption of Sirt3/angiopoietins/Tie-2 and HIF-2α/Notch3 pathways is also critical for LPS-induced lung pericyte loss [141]. Importantly, pericyte transplantation has been shown to reduce pericyte loss and increase the survival rate in septic rats [142]. In addition, microvesicles derived from pericytes have improved pulmonary function in a rat model of sepsis [145]. These findings suggest that therapeutic strategies targeting pericytes for sepsis hold promise, and a further understanding of the underlying mechanisms of pericyte dysfunction and loss in sepsis is needed. 

#### 2.3.2. Pericyte Loss in HIV

The neurocognitive disorder is a major complication of HIV as the virus enters the brain shortly after infection, leading to inflammation and BBB disruption [146,147]. In vitro studies have demonstrated that cultured brain pericytes can be infected by HIV, resulting in enhanced production of inflammatory mediators and disruption of endothelial barrier properties [148,149]. Furthermore, evidence from in vivo studies, including HIV patients and mouse models of HIV, has shown that brain pericytes can be infected by HIV [150,151,152]. Following HIV infection, a reduction in pericyte coverage has been observed in the brains of HIV patients [150,153,154]. Similar pericyte loss has also been detected in the brains of mouse models of HIV and SIV-infected macaques [153,154]. It has been suggested that the higher concentration of PDGF-BB induced by HIV Tat via the activation of mitogen-activated protein kinases and nuclear factor-κB pathways may drive HIV-induced pericyte loss in the brain [153,155]. However, the role of pericytes in HIV has not been extensively examined. A better understanding of pericyte dysfunction and loss in the context of HIV may provide opportunities for the development of novel therapeutics.

#### 2.3.3. Pericyte Loss in COVID-19

COVID-19 is caused by the severe acute respiratory syndrome coronavirus 2 (SARS-CoV-2) and affects various organs, including the heart, brain, and lungs [156,157,158,159]. Cardiac pericytes, which express high levels of angiotensin-converting enzyme 2 (ACE-2), the main receptor for SARS-CoV-2, are major targets for viral infection [160,161,162]. The infection of pericytes via SARS-CoV-2 contributes to cardiac complications associated with COVID-19, such as thrombosis, inflammation, and hemodynamic disturbances [163]. Studies have shown a significant loss of pericyte coverage in the heart capillaries of hamsters infected with SARS-CoV-2 [164]. Additionally, SARS-CoV-2 can infect cardiac pericytes, and its spike protein may induce pericyte dysfunction via CD147 receptor-mediated signaling pathway, leading to microvascular injury [157]. Brain pericytes, which also express ACE-2, are susceptible to SARS-CoV-2 infection, potentially driving inflammation and vascular dysfunction [158,165,166]. Patients with COVID-19 have shown lower levels of the pericyte marker PDGFRβ in their cerebrospinal fluid [158]. SARS-CoV-2 spike protein has been found to deregulate vascular and immune functions in brain pericytes [167], while the SARS-CoV-2 envelope protein has been shown to induce brain pericyte death in vitro [168]. In the lung, pericytes were infected by SARS-CoV-2 and are detached from pulmonary capillary endothelium in COVID-19 patients [159,169]. However, the underlying mechanisms of pericyte loss in COVID-19 remain largely unknown, and further studies are needed to investigate the role of pericytes, particularly in long COVID-19 [170]. 

Overall, the loss of pericytes in infectious diseases can have significant negative effects on patient outcomes. Understanding the mechanisms underlying pericyte loss in these diseases is crucial for the development of new treatments that can prevent or reverse this process and improve patient outcomes. Further research is needed to gain a better understanding of the role of pericytes in infectious diseases and to explore novel therapeutic approaches that can target these cells.

### 2.4. Pericyte Loss in Brain Injury

#### 2.4.1. Pericyte Loss in Stroke

Stroke, a leading cause of death and disability worldwide, is associated with pericyte dysfunction and BBB disruption [171]. Pericytes play a critical role in regulating inflammation, angiogenesis and BBB function during stroke [171,172]. A rapid reduction in brain pericyte number and coverage has been observed in human stroke cases as well as in experimental stroke models, including mice and rats, following ischemic damage [113,173,174,175,176]. Pericyte apoptosis and autophagy have been detected in the brain from murine models of stroke, which may contribute to pericyte loss and BBB disruption [174,175]. The loss of regulator of G protein signaling 5 (RGS5) has been associated with increased pericyte number and improved BBB function in a mouse model of stroke, suggesting a role for RGS5 in brain pericyte loss during stroke [177]. Additionally, the inhibition of Sema3E/PlexinD1 signaling has been shown to increase pericyte number and enhance blood–brain barrier integrity in aged rats with stroke, further implicating this signaling pathway in brain pericyte loss [176]. Moreover, the deletion of hypoxia-inducible factors (HIF)-1 in pericytes has been found to prevent brain pericyte apoptosis and reduce vascular permeability in mice with stroke, indicating the involvement of pericyte HIF-1 in stroke-induced pericyte apoptosis [175]. In addition to maintaining BBB function by themselves, pericytes also promote the physiological functions of other BBB components including endothelial cells, basal lamina, and astrocytes [178]. For example, pericytes regulate aquaporin-4 polarization in mouse cortical astrocytes [179]. Furthermore, angiopoietin-1 secreted by pericytes mediates tight junction induction via the activation of Tie-2, an angiopoietin-1 receptor on EC [178,180,181]. Therefore, restoration of pericyte coverage may improve BBB support and promote reperfusion after stroke [182]. Gaining more insights into the role of pericytes in stroke could facilitate the development of novel therapeutic approaches for stroke treatment [172].

#### 2.4.2. Pericyte Loss in Traumatic Brain Injury (TBI)

TBI, caused by an external force, is the major cause of mortality and disability, particularly in young individuals [183]. The secondary injury following TBI involves oxidative stress, inflammation, and the production of matrix metalloproteinases (MMPs), which contribute to BBB dysfunction [184,185,186]. Recent studies have highlighted pericyte degeneration as a significant factor in TBI, leading to regional microcirculatory hypoperfusion and increased BBB permeability [187,188]. A decline in pericyte markers has been observed in brain specimens from human TBI cases and in a mouse model of repetitive mild TBI up to 12 months post-injury [189]. Additionally, rapid pericyte loss in the acute phase of TBI has been documented in the brains of mice with TBI [188,189,190,191,192]. Brain pericyte apoptosis has been detected in a mouse model of TBI, suggesting that pericyte loss during TBI may be attributed to apoptosis [193]. It has been found that the inhibition of the TNF-α/NF-κB/iNOS axis can reverse pericyte loss, improve pericyte function, and enhance microcirculation perfusion after TBI [188]. This indicates the potential contribution of the TNF-α/NF-κB/iNOS axis to pericyte loss in TBI. Consequently, the development of treatments that can prevent or reverse pericyte degeneration holds promise for the management of TBI and the secondary injuries that follow.

Overall, understanding the mechanisms of pericyte loss in these conditions is crucial for developing new treatments that can prevent or reverse this process and improve patient outcomes. 

## 3. Methods to Determine Pericyte Loss

As there is no universally recognized marker for pericytes, and pericyte markers can vary depending on the specific tissues, it becomes crucial to employ various methods for detecting and assessing pericyte changes. In this section, we will explore several approaches commonly used to determine and evaluate alterations in pericytes.

### 3.1. Immunohistochemistry

One commonly utilized method to assess pericyte changes is immunohistochemistry, which enables the visualization and quantification of pericytes within tissues. This technique involves staining tissue samples with specific antibodies targeting pericyte markers and/or vascular markers (Table 2). In studies focusing on the brain, pericytes have been identified via the immunostaining of various markers, such as PDGFR-β [95,97,118,191], CD13 [7], NG2 [97], NG2+CD31 [153], CD13+CD31 [150], PDGFR-β+CD31 [194], a-SMA+laminin [138], PDGFR-β+laminin [195], PDGFR-β+lectin [196], CD13+lectin [119,192], and desmin+lectin [197]. In retinal investigations, pericytes have been detected via the immunostaining of markers such as α-SMA [42], NG2 [33,38,42], and PDGFR-β+lectin [117,120] allowing for the determination of pericyte number and coverage. Lung pericytes have been characterized by the immunostaining of NG2+IB4 [141] and Foxd1+CD31 [15]. Similarly, cardiac pericyte changes have been observed via the immunostaining of NG2 [108], NG2+IB4 [24,141], NG2+isolectin [164], and NG2+PDGFR-β [157]. In the case of kidneys, immunostaining with antibodies specific to WT1 [78,81,82] has been employed to detect changes in renal podocytes, which are pericyte-like cells in the kidneys. Additionally, immunohistochemical staining of CD13+laminin has been utilized to assess pericyte changes in the spinal cord [133]. Collectively, combining pericyte markers with vascular markers remains the predominant approach for immunohistochemical detection of pericyte density and coverage in various tissues.

### 3.2. Electron Microscopy

Another valuable method utilized for assessing pericyte loss is electron microscopy. This technique involves capturing high-resolution images of tissue samples using an electron microscope, enabling the visualization of cellular structures with great detail. Electron microscopy is particularly effective in visualizing pericytes and detecting alterations in their morphology, such as detachment from blood vessels, shrinkage, or loss of cellular organelles. For instance, in studies involving diabetic mice, transmission electron microscopy (TEM) has been employed to detect changes in the morphology and density of renal podocytes [76]. Scanning electron microscopy has been utilized to observe the detachment of brain pericytes from the capillary wall [198]. Additionally, TEM has proven useful in visualizing the interaction between pericytes and endothelium [199]. Furthermore, TEM has been widely applied to identify pericyte changes in various tissues affected by different diseases [174,200,201,202].

### 3.3. Live Animal Imaging Techniques

Live animal imaging techniques, such as intravital microscopy or two-photon microscopy, play a crucial role in determining pericyte changes in vivo. These advanced methods allow for the visualization of live tissues using fluorescently labeled antibodies or cells. By employing live animal imaging, researchers gain real-time, dynamic information about pericyte behavior and their interactions with other cells. For instance, confocal intravital microscopy has been successfully utilized to study the interactions between neutrophils and pericytes in vivo [203]. Furthermore, intravital microscopy has been employed to investigate the dynamic interactions of endothelial cells and pericytes [204,205]. Changes in pericytes can be detected via alterations in fluorescent signals or by observing modifications in the morphology or behavior of fluorescently labeled cells. Two-photon microscopy has also been widely used to study dynamic changes, spatial distribution, density, and the subsets of pericytes [206,207,208,209,210]. These live animal imaging techniques are invaluable for investigating the role and mechanisms of pericytes in regulating vascular function. Additionally, they provide crucial insights into the dynamic interactions between pericytes and surrounding cells during different stages of diseases. These techniques hold great promise for future research endeavors in this field. 

### 3.4. Other Techniques

Additional techniques, such as Western blot and ELISA, have been employed as supportive methods to determine pericyte changes by quantifying protein levels of pericyte markers in tissues and cerebrospinal fluid (CSF). Western blot analysis has revealed decreased expression levels of pericyte markers in the lung of septic mice, corroborating the observations of lung pericyte loss reported in immunohistochemistry studies [15,141]. ELISA measurements of PDGFR-β expression have shown correlations with pericyte numbers in the brain white matter of stroke and Alzheimer’s disease patients [113]. Furthermore, ELISA-based detection of soluble PDGFRβ levels in the CSF serves as an indicator of brain pericyte injury, often associated with blood–brain barrier breakdown [211,212,213]. Recent advancements in molecular biology techniques, such as single-cell sequencing and RNA sequencing, have revolutionized the identification and characterization of pericytes at the molecular level. These cutting-edge techniques offer valuable insights into the gene expression patterns of pericytes, the subsets of pericytes, and how they undergo changes in response to diseases or injuries [214,215,216,217,218]. By employing these techniques, researchers can identify subtypes of pericytes and discern their distinct roles in various diseases. 

In summary, the detection of pericyte loss is crucial for the diagnosis and treatment of numerous diseases. There are several methods available to determine pericyte changes, including immunohistochemistry, electron microscopy, live imaging, and molecular biology techniques. The selection of the appropriate method depends on the specific research question, the type of tissue being studied, and the availability of resources. Combining multiple techniques can offer a more comprehensive understanding of the role and dynamic changes of pericytes in different disease contexts. By employing a multidimensional approach, researchers can gain valuable insights into the complex behavior and functions of pericytes, facilitating advancements in disease diagnosis, treatment, and therapeutic interventions.

## 4. Conclusions

In conclusion, the detection of pericyte loss or dysfunction has been established in various diseases, and their contribution to pathological progression is well recognized. Pericytes exhibit multifunctional properties, offering potential avenues for therapeutic interventions in conditions involving inflammation, fibrosis, angiogenesis, and vascular dysfunction [171,182,219]. Preclinical studies have demonstrated the efficacy of treatments targeting pericytes, such as modulating gene expression or implanting pericytes, in animal models of sepsis [15], stroke [175], diabetic retinopathy [53], Alzheimer’s disease [124], and amyotrophic lateral sclerosis [134]. Future therapeutic approaches targeting pericytes/pericyte-like cells can be explored from several of the following angles: (1) modulation of signaling pathways in pericytes or surrounding cells that contribute to pericyte loss or dysfunction; (2) reduction in detrimental factors that induce damage and degeneration in pericytes; (3) implementation of pericytes or specific subpopulations derived from various organ origins for cell-based therapies; (4) utilization of multipotential stem cells to generate pericytes/pericyte-like cells for implantation; (5) utilization of exosomes derived from healthy or modified pericytes. Taken together, pericytes represent a promising target for the development of novel therapeutic treatments. Further research and advancements in understanding pericyte biology and its interactions within the microenvironment will enhance our ability to harness the therapeutic potential of pericytes, leading to improved clinical outcomes in a wide range of diseases.

## Figures and Tables

**Table 1 cells-12-01931-t001:** The function and coverage of pericytes in organs.

Tissue	Function	Pericytes/EC Ratio
Brain	Maintaining BBB function, the recruitment of inflammatory cells, regulating cerebral blood flow, Aβ clearance, and inflammation	1:1
Retina	Maintaining BRB function, regulating Aβ clearance, and inducing immune responses	1:1
Lung	Regulating inflammatory response, maintaining the pulmonary vasculature, and optimal gas exchange	1:7–1:9
Kidney	Maintaining the integrity of peritubular capillaries, regulating blood ultrafiltration and vascular permeability	1:2.5
Liver	Remodeling of the ECM, vitamin A metabolism, and the recruitment of inflammatory cells	1:10
Heart	Regulating blood flow, vascular remodeling, and myocardial and interstitial fibrosis	1:2–1:3

Aβ: amyloid β-peptide; BBB: Blood-brain barrier; BRB: blood–retinal barrier; EC: endothelial cells; ECM: extracellular matrix.

**Table 2 cells-12-01931-t002:** Markers used to detect pericytes in different tissues via immunohistochemistry.

Tissue	Disease	Pericyte and/or Vascular Markers
Brain	Diabetes	PDGFR-β [95,97], NG2 [97]
Brain	Sepsis	a-SMA+laminin [138]
Brain	HIV	NG2+CD31 [153], CD13+CD31 [150]
Brain	AD	CD13 [7], PDGFR-β [118], CD13+lectin [119], PDGFR-β+lectin [196]
Brain	MS	PDGFR-β+ laminin [195], Desmin+lectin [197]
Brain	Brain metastases	PDGFR-β+CD31 [194]
Brain	TBI	PDGFR-β [191], CD13+lectin [192]
Retina	Diabetes	a-SMA [42], NG2 [33,38,42]
Retina	AD	PDGFR-β+lectin [117,120]
Lung	Sepsis	NG2+IB4 [141], Foxd1+CD31 [15]
Heart	Diabetes	NG2 [108], NG2+IB4 [24]
Heart	Sepsis	NG2+IB4 [141]
Heart	COVID-19	NG2+ isolectin [164], NG2+ PDGFR-β [157]
Kidney	Diabetes	WT1 [78,81,82]
Spinal cord	ALS	CD13+laminin [133]

AD: Alzheimer’s disease; MS: multiple sclerosis; TBI: traumatic brain injury; ALS: amyotrophic lateral sclerosis.

## Data Availability

No new data were created.

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
