# Peer review of "Pericyte Loss in Diseases"

_cells, 2023, doi:10.3390/cells12151931_

Round 1

Reviewer 1 Report (Previous Reviewer 2)

Accept in its current form.

Author Response

We appreciate the positive feedback from the reviewer.

Reviewer 2 Report (New Reviewer)

In few words , I have really appreciated the review entitled "Pericyte loss in diseases" by Pengfei Li, Hongkuan Fan. As usual , this review is in some parts difficult to read due to the tremendous amount of data  presented. More in depth, pericyte loss seems to be associated with metabolic alterations/inflammation/degeneration as a universal marker of tissue and vessels impairment. I have not specific criticisms to do but, due to this general behavior of  pericyte loss in tissues, the possible starting insult should be presented at glance in the paper. For this reason, I recommend to add some figures to show the possible initial alterations that like an etiologic factor induce tissue damage and consequent pericyte loss. 

no comments to do

Author Response

We appreciate the positive feedback from the reviewer. We would like to highlight that our current understanding of the role and specific mechanisms of pericyte loss in relevant tissues is still limited and largely unknown. Additionally, it is important to recognize that the mechanisms and patterns of pericyte loss can vary across different diseases, even within the same tissue. According to the reviewer’s insightful suggestion, we propose including a figure abstract that summarizes the key points of this review as below. This figure abstract can provide a visual representation of the main findings and concepts discussed in the article, helping to enhance the overall understanding and impact of the review.

Reviewer 3 Report (New Reviewer)

Pericytes are the most fascinating cells among the heterogeneous population of   stromal lineage cells.  They got their name exclusively because of their anatomical location - around the endothelium, inside its basement membrane. However, these cells do not yet have specific molecular markers. Their intimate association with the vascular bed is the determinant of their biological role. Like endothelial tubes, the pericyte network permeates all tissues of the body. Therefore, the negative consequences of pericyte loss in various pathologies is a very intriguing issue, connected with the consequences not only for local but also systemic response of the organism.   Thus, the scientific importance of MS is not left to doubt.

 In the introduction, information not only on the immunophenotype of pericytes, but also on their morphology in the context of their mesenchymal origin and histogenesis relevance would be appreciated.

L81 "Therefore, pericyte-targeted therapies or cell implantation of pericytes".  

If pericytes are considered strictly according to their position within the basal membrane of endothelium, then how can cells from the stromal-vascular fraction, which is represented by a mixture of cells of the stromal lineage, be considered as pericytes and be applied for a pericyte-based therapy? If pericytes do not have definitive markers, how can they be selected for therapy? Authors should clearly distinguish between information that is specific to pericytes and knowledge of pericyte-like cells.

L88 "Renal pericytes, including peritubular pericytes, mesangial cells and podocytes".

This phrase is a mixture of pericytes by definition, mesangial cells that are also present in the matrix of the glomeruli, and cells of epithelial origin - podocytes.

The authors have provided detailed information on the actual loss of pericytes in a variety of pathological conditions. Nevertheless, the review would be significantly strengthened with the data on the reaction of pericytes local milieu (i.e. endothelial lining, etc.) on the loss of pericytes.  How the loss of pericytes in a certain location is reflected on the state of the capillary network in other tissues, how is compensated, if so.

Author Response

Reviewer 3

Pericytes are the most fascinating cells among the heterogeneous population of   stromal lineage cells.  They got their name exclusively because of their anatomical location - around the endothelium, inside its basement membrane. However, these cells do not yet have specific molecular markers. Their intimate association with the vascular bed is the determinant of their biological role. Like endothelial tubes, the pericyte network permeates all tissues of the body. Therefore, the negative consequences of pericyte loss in various pathologies is a very intriguing issue, connected with the consequences not only for local but also systemic response of the organism.   Thus, the scientific importance of MS is not left to doubt.

 In the introduction, information not only on the immunophenotype of pericytes, but also on their morphology in the context of their mesenchymal origin and histogenesis relevance would be appreciated.

Response: We appreciate the reviewer’s insightful suggestion and have included the morphology and histogenesis relevance of the pericytes as far as we know in the Introdcution part of the revised manuscript.

L81 "Therefore, pericyte-targeted therapies or cell implantation of pericytes".  

If pericytes are considered strictly according to their position within the basal membrane of endothelium, then how can cells from the stromal-vascular fraction, which is represented by a mixture of cells of the stromal lineage, be considered as pericytes and be applied for a pericyte-based therapy? If pericytes do not have definitive markers, how can they be selected for therapy? Authors should clearly distinguish between information that is specific to pericytes and knowledge of pericyte-like cells.

Response: We appreciate the reviewer’s insightful suggestion. We have corrected this sentence and other similar sentences in the revised manuscript.

L88 "Renal pericytes, including peritubular pericytes, mesangial cells and podocytes".

This phrase is a mixture of pericytes by definition, mesangial cells that are also present in the matrix of the glomeruli, and cells of epithelial origin - podocytes.

The authors have provided detailed information on the actual loss of pericytes in a variety of pathological conditions. Nevertheless, the review would be significantly strengthened with the data on the reaction of pericytes local milieu (i.e. endothelial lining, etc.) on the loss of pericytes.  How the loss of pericytes in a certain location is reflected on the state of the capillary network in other tissues, how is compensated, if so.

Response: We appreciate the reviewer’s insightful suggestion. It is important to note that the primary objective of this review is to provide an updated summary of pericyte loss across a wide range of diseases, while also emphasizing the significant features and potential mechanisms associated with each disease. In addition, the data on the reaction of pericytes local milieu (i.e. endothelial lining, etc.) on the loss of pericytes, how the loss of pericytes in a certain location is reflected on the state of the capillary network in other tissues, and how is compensated are still largely unknown. Despite this limitation, we have made significant efforts to incorporate as much detail as possible in the revised article.

This manuscript is a resubmission of an earlier submission. The following is a list of the peer review reports and author responses from that submission.

Round 1

Reviewer 1 Report

Increasing studies reveal the vital role of pericyte in maintaining tissue homeostasis.  In this review, the authors provided a timely update on the important roles of pericyte loss during the pathogenesis of relevant diseases.  These timely perspectives are useful for the field of pericyte biology related to disease.  In addition, the authors provided authoritative descriptions of important methodologies in field of pericyte examination including immunohistochemistry, live animal imaging, and other related techniques, very helpful to the field.  The review should be accepted in its current form, with some minor suggestion as below: 

The authors should consider adding a figure illustration depicting the pericyte localization within relevant tissues (e.g. vasculature, brain, etc) responsible for their vital roles, and how pericyte loss may lead to tissue leakage, would strengthen the readability of the review. 

Author Response

We appreciate the positive feedback from the reviewer. We would like to highlight that our current understanding of the role and specific mechanisms of pericyte loss in relevant tissues is still limited and largely unknown. Additionally, it is important to recognize that the mechanisms and patterns of pericyte loss can vary across different diseases, even within the same tissue. As a result, incorporating a figure illustration depicting the localization of pericytes within relevant tissues and their specific functions would be challenging at this stage.

Instead of adding a figure in the article, we propose including a figure abstract that summarizes the key points of this review. This figure abstract can provide a visual representation of the main findings and concepts discussed in the article, helping to enhance the overall understanding and impact of the review.

Reviewer 2 Report

Though the functions of pericytes are suggested to be involved in pathogenesis of various diseases, it is highly likely that the causal relationship is not always clear in every disease entity. We can expect that disease progression can determine mobilization and function of pericytes. Also, pericyte is a group of many different cells with various origins and possibly different functions.

The authors are recommended to focus on limited diseases where animal studies confirmed the role of pericytes in each disease progression and to add references.

Author Response

We appreciate the positive feedback provided by the reviewer. In response to the comments, we have made significant revisions to the article, particularly by incorporating additional details on diseases where animal studies have confirmed the role of pericytes in disease progression. These additions aim to provide a more comprehensive understanding of the involvement of pericytes in specific diseases and enhance the overall quality and relevance of the article.

Reviewer 3 Report

Withing this review the authors try to discuss the topic of pericyte loss in many different diseases. Overall I think that the review gives a lot of useful information but it tries to cover too many different things without giving enough details for any of them. I would suggest to elaborate more on the single paragraphs.

Specific comments:

I think table 1 should be more detailed and should actually address all the specific functions of pericytes. Maintaining the integrity of the BBB/BRB is a bit too general, given the vital role that pericytes have in the brain and retina.

The structure of the review is a bit confusing as it starts by addressing pericyte loss associated with specific diabetes-related diseases but than it shifts to pericyte loss in specific organs. I would also write a general paragraph about pericyte loss in diabetes before going into details with the diabetes-realated diseases.

I think brain injury and brain metastases should be separated paragraphs as they are completely unrelated.

Overall I think the paragraph should be expanded to give a more detailed and complete picture of the different diseases and the signalling pathways involved. 

No issue with that.

Author Response

Specific comments:

I think table 1 should be more detailed and should actually address all the specific functions of pericytes. Maintaining the integrity of the BBB/BRB is a bit too general, given the vital role that pericytes have in the brain and retina.

Response: We greatly appreciate the reviewer's insightful comment. In response, we have made the necessary revisions to the article, specifically by including the specific functions of pericytes in Table 1.

The structure of the review is a bit confusing as it starts by addressing pericyte loss associated with specific diabetes-related diseases but than it shifts to pericyte loss in specific organs. I would also write a general paragraph about pericyte loss in diabetes before going into details with the diabetes-realated diseases.

Response: We sincerely thank the reviewer for the insightful suggestion. In response to the feedback, we have included a dedicated paragraph in the revised article discussing the topic of pericyte loss in diabetes.

I think brain injury and brain metastases should be separated paragraphs as they are completely unrelated.

Response: We thank for the reviewer’s suggestion. We have separated the brain injury and brain metastases in the revised article.

Overall I think the paragraph should be expanded to give a more detailed and complete picture of the different diseases and the signalling pathways involved.

Response: We appreciate the reviewer's suggestion. It is important to note that the primary objective of this review is to provide an updated summary of pericyte loss across a wide range of diseases, while also emphasizing the significant features and potential mechanisms associated with each disease. We acknowledge that in certain diseases, such as diabetes, there are extensive discussions on the detailed picture and involved signaling pathways related to pericyte loss in various review articles. However, it is worth noting that in many other diseases, including HIV, Covid-19, amyotrophic lateral sclerosis, multiple sclerosis, tuberculosis, and malaria, the signaling pathways associated with pericyte loss are still largely unknown. Despite this limitation, we have made significant efforts to incorporate as much detail as possible in the revised article, providing a comprehensive overview of the current understanding and available information on pericyte loss in these diseases.

Round 2

Reviewer 2 Report

The authors did not introduce any significant change in response to my critiques before. Neither the manuscript has been significantly improved to warrant publication in cells, nor the authors directly responded to my questions at all. 

Reviewer 3 Report

Thank you for addressing my comments. The review has improved and it is now ready for publication.

No major issue detected.